# Transcription factor competition facilitates self-sustained oscillations in single gene genetic circuits

**Jasper Landman**[1]*, **Sjoerd M. Verduyn Lunel**[2], **Willem K. Kegel**[3]

**1** Physics & Physical Chemistry of Foods, Wageningen University & Research, Wageningen, the Netherlands, **2** Mathematisch Instituut, Utrecht University, Utrecht, the Netherlands, **3** Van 't Hoff Laboratory for Physical & Colloid Chemistry, Utrecht University, Utrecht, the Netherlands

* jasper.landman@wur.nl

**Data Availability Statement:** All relevant code is provided in the supplementary materials of this article. See S2 Appendix used to generate all the graphs in the paper (S2 Appendix).

## Abstract

Genetic feedback loops can be used by cells to regulate internal processes or to keep track of time. It is often thought that, for a genetic circuit to display self-sustained oscillations, a degree of cooperativity is needed in the binding and unbinding of actor species. This cooperativity is usually modeled using a Hill function, regardless of the actual promoter architecture. Furthermore, genetic circuits do not operate in isolation and often transcription factors are shared between different promoters. In this work we show how mathematical modelling of genetic feedback loops can be facilitated with a mechanistic fold-change function that takes into account the titration effect caused by competing binding sites for transcription factors. The model shows how the titration effect facilitates self-sustained oscillations in a minimal genetic feedback loop: a gene that produces its own repressor directly without cooperative transcription factor binding. The use of delay-differential equations leads to a stability contour that predicts whether a genetic feedback loop will show self-sustained oscillations, even when taking the bursty nature of transcription into account.

## Author summary

Cells keep track of time by genetic feedback loops—transcription factor proteins that repress their own production and whose availability oscillates regularly in time. For oscillations to occur, the amount of transcription factor that is produced should be extremely sensitive to the amount of transcription factor that is present in the cell. In some genetic circuits, sensitivity comes from the fact that multiple transcription factors need to bind to the gene simultaneously to successfully inhibit its production, but it is also possible to generate this sensitivity when multiple binding sites compete for the same pool of transcription factors. In this work we show how to mathematically model this competition effect, and predict under which circumstances stable oscillations occur.

**Funding:** The author(s) received no specific funding for this work.

**Competing interests:** The authors have declared that no competing interests exist.

## Introduction

The 2017 Nobel prize in Chemistry was awarded jointly to Michael Young, Michael Rosbash and Jeffrey Hall for their work on molecular mechanisms controlling the circadian rhythm [1–3], highlighting the importance of oscillating reactions to living cells. The cellular analogue of classical clock reactions like the Belousov-Zhabotinsky reaction [4, 5], self-sustained oscillations require a network of reactions incorporating a negative feedback loop [2, 6–16]. Such oscillatory networks can be used to coordinate important processes in the cell, for example, cell division [8, 11, 13, 17–20]. Oscillatory circuits may include both enzymatic binding and unbinding events, as well as transcriptional elements. However, the majority of cellular oscillatory reactions that have been observed incorporate a transcription event in the feedback loop [21] allowing the regulation of parallel transcription processes. Progress in synthetic biology has made it possible to design circuits of transcriptional elements that perform desired functions within a cell (see, e.g., [22–25]) or even across multiple organisms [26].

According to Novak and Tyson [21], sustained oscillations in networks require the following three ingredients: first of all, the presence of a negative feedback loop. Without a negative feedback loop a system can not be driven back to its original state. The second requirement is a time delay in the feedback loop. When the feedback is instantaneous, any perturbation of the system can be driven back to its steady-state. Finally, it is required to have a certain degree of cooperativity. In models, this cooperativity assumption is usually modelled using a Hill function [27] in the binding probability of a transcription factor. The Hill equation has the following form

$$\theta = \frac{[L]^n}{K_d^n + [L]^n}, \qquad (\text{Hill equation}) \qquad (1)$$

where $\theta$ is the occupation fraction of an adsorbed ligand to a lattice site, $[L]$ is the concentration of ligand (monomer), $K_d$ the dissociation constant and $n$ the Hill exponent (the right hand side of this equation is called the Hill function). For $n = 1$, the Hill equation is equal to the Langmuir isotherm. The Hill equation was postulated for the cooperative binding of oxygen to hemoglobin [27], modelling it as the simultaneous binding of $n$ ligands to a lattice site. However, often in literature, a Hill equation is used to describe the binding of a ligand with a different binding architecture. In those cases, an 'effective' Hill exponent is used as a fitting parameter and as a measure for the cooperativity of binding. Hill himself admitted that "[his] object was rather to see whether an equation of *this type* can satisfy all the observations, than to base any direct physical meaning on $n$ and $K$" [27]. The consequence of a higher effective Hill exponent on the binding isotherm of a ligand is a very sharp transition from mostly free ligand to mostly bound ligand. It is this steep response that leads to oscillations.

Often, a transcription factor involved in an oscillatory circuit is shared by a number of other genes. The circuit and the regulated gene are effectively competing for a common pool of transcription factors [28–30] which consequently leads to a titration effect—A dramatic increase in transcription factor occupancy is observed when the copy number of transcription factors crosses over from a regime where transcription factors are limiting to a regime of transcription factor excess. The increase is similar to the sharp transition found in the cooperative binding of ligands, as illustrated in Fig A in S1 Appendix.

Although the notion exists that a competition-driven titration effect can also lead to sufficient nonlinearity to generate self-sustained oscillating reactions—such circuits are coined 'titration oscillators' [31–34]—equilibrium modelling of transcription factor binding in these circumstances has so far resorted to the use of mechanistically incorrect Hill equation. [35–40] Kim and Tyson [41] level similar criticism at the use of quasi-Steady State Michaelis Menten

kinetics which also reduces to a Hill equation. Nonequilibrium modelling of titration oscillators can lead to mechanistically correct results, at the cost of requiring more parameters.

In this article we will show how the rate equations that govern the behaviour of genetic circuits lead to a very natural inclusion of previously derived results that allow a mechanistic integration of the response function of a gene [28–30]. Using the assumption that transcription and translation are slow processes in comparison to the binding and unbinding kinetics of transcription factors, we can use expressions for the fold-change of genes derived in the grand canonical ensemble, which account for the presence of other binding sites, such as multiple gene copies, competitor sites and inhibitors, competing for the same transcription factor, and for complex regulatory architectures with multiple binding sites. We show that the presence of a small number of competitor sites or inhibitors within the cell generates sufficient nonlinearity in the response curve of the gene to allow self-sustained oscillation in a minimal genetic feedback loop: a gene that produces its own repressor.

We use delay differential equations (DDEs) [42] to incorporate in the model the finite time taken by the transcriptional and translational process. DDEs allow us to relate the change in mRNA copy number to the concentration of transcription factors at some time in the past. We restrict ourselves to DDEs with constant time delays and do not include the stochastics of transcription and translation explicitly. For that, a master equation approach would be required. We show, however, that one can account for variances in the time delay implicitly to evaluate the stability of an oscillating circuit. DDEs were previously used in the context of oscillations in genetic circuits, for example in Smith [43]. We should note that Korsbo and Jönsson [44] are critical of the use of DDEs within the context of modelling transcriptional regulation, and they provide an alternative approach based on a constant rate of information propagation along a variable path length. Another approach, taken by Bratsun *et al.* [45], involved integration of genetic circuits using a delay-modified Gillespie algorithm [46]. In this paper, stochastic perturbations are taken into account explicitly, although their discussion limits itself to simple promoter architectures in isolation, for which the Hill equation holds.

While constant delay DDEs may not capture the full behaviour of this proposed alternative, the use of DDEs in modelling provides a much more realistic theory than can be given using ordinary differential equations, and allows a full stability and bifurcation analysis using the time delay as a bifurcation parameter [42]. Furthermore, as a result of our analysis, we can show that for a single membered genetic circuit it is possible to derive a criterion that predicts whether self-sustained oscillations are possible. The criterion just depends on the slope of the fold-change curve of the promoter architecture, and on the time delay that is inherent in the transcription and translation process.

Finally we take a closer look at an experimental example from the work of Stricker *et al.* [47], who have shown oscillations in LacI concentration in a genetic circuit with only a single negative feedback loop, when LacI is induced by IPTG. Here, we show that if the IPTG inducer is modelled as a competitive binding site for the repressive transcription factor, then our model indeed predicts oscillations with similar features as observed in the experiment.

## Methods

### Gene regulation in the grand canonical ensemble

Since the seminal work of Von Hippel *et al.* [48], equilibrium based models have been used successfully to quantitatively predict the transcriptional activity of genes in the presence of transcription factors. These models are all based on the same cornerstone: that transcriptional activity—an inherently non-equilibrium process—can be taken as proportional to the (equilibrium) probability that their promoter region is occupied by RNA polymerase (RNAP), an

assumption which is justified in case the formation of the RNAP open complex on the promoter site of a gene is slow in comparison to the binding and unbinding kinetics of transcription factors over the genome [49–52]. Under these assumptions, equilibrium statistical mechanics can be used to calculate the RNAP occupancy. The assumptions needed to treat transcription regulation as a quasi-equilibrium process are subtle (see, e.g., ref [53, 54], and there exists a corresponding class of kinetic models, which do not require as many assumptions, at the cost of requiring more parameters [55–61].

Statistical mechanics [62] provide the tools needed to calculate the occupation probability of RNAP and transcription factors to binding sites on the DNA. In this approach, transcription initiation is modeled as a lattice on which RNAP and transcription factors occupy either their cognate binding sites, or empty non-specific sites. The transcriptional activity is then taken as proportional to the equilibrium occupational probability of RNAP to the gene promoter sequence—a precondition for the subsequent initiation of the transcription process.

If we assume that only a single specific binding site is available to a transcription factor, then its distribution can be calculated directly from the canonical partition sum of the genome [63]. Different promoter architectures on the gene of interest lead to different configurational states, each with their own statistical weight [64, 65]. However, when multiple specific binding sites are available to a transcription factor, keeping track of the combinatorics that describe all the different possible configurational states quickly becomes tedious.

In previous work [29, 30] we have developed an alternative approach to account for the presence of multiple specific binding sites, such as multiple gene copies or different genes that compete for a shared pool of transcription factors. Rather than considering the entire genome as a canonical ensemble, we consider only the gene of interest as a grand canonical ensemble, coupled to one or more reservoirs. In a grand canonical ensemble, the number of particles is allowed to fluctuate, but their chemical potential $\mu$ is fixed. Consequently, in this setup the effect of competing binding sites for transcription factors are taken into account indirectly, as the different reservoirs are coupled only through the chemical potential of the transcription factor.

The grand canonical ensemble formulation is particularly well suited to deal with systems where many species can bind to different sites. In principle, the models we propose in this work can also be derived from the more conventional biochemical formulation based on multiple chemical equilibria and reaction kinetics, such as presented in the work of e.g., Klumpp and Hwa [66], at the cost of requiring more parameters. We have chosen the grand canonical ensemble formulation, as it makes the link with a mechanistic equilibrium adsorption problem obvious.

The grand canonical partition sum [62] of a single gene copy is found by summing over the possible occupation numbers of all proteins $i$ with a specific binding site within the promoter architecture of the gene. By convention, we give RNA Polymerase the index 1. The grand canonical partition sum then reads

$$\Xi = \sum_{p_1, p_2, \cdots} \lambda_P^{p_1} \lambda_P^{p_2} \cdots Z(p_1, p_2, \ldots),$$

(2)

Here the indices of the sum $p_1, p_2, \ldots$ are the occupancies of proteins 1, 2, ... on the promoter architecture. Each protein $i$ has a fugacity $\lambda_i = \exp(\beta\mu_i)$ with $\mu_i$ its chemical potential. The factor $\beta = (k_B T)^{-1}$ is the inverse thermal energy, with $k_B$ Boltzmann's constant, and $T$ the absolute temperature. Furthermore, $Z(p_1, p_2, \ldots)$ is the relevant part of the canonical partition sum of the gene copy with $(p_1, p_2, \ldots)$ copies of regulating transcription factor adsorbed. As such, the factor $Z = \exp(-\beta F)$ is related to the change in the Helmholtz free energy upon binding $(p_1, p_2,$

. . .) copies of regulating transcription factor. When $N$ multiple copies of the same gene are present, assuming all copies are independent, the grand canonical partition function of all copies is $\Xi_N = \Xi^N$.

Finding the grand canonical partition sum of a gene of interest is rather straightforward: one simply needs to sum the statistical weights of all possible configurational states. The possible states depend on the promoter architecture of the gene of interest, and can usually be written out by hand.

From the grand canonical partition sum, we can calculate the expected fraction of specific binding sites occupied by its cognate transcription factor through

$$\theta_i = \frac{\langle p_i \rangle}{N} = \frac{\lambda_i}{\Xi}\left(\frac{\partial \Xi}{\partial \lambda_i}\right). \tag{3}$$

The fugacity of the regulatory transcription factor is set by the copy number of that transcription factor and the genetic environment. For each type of binding site that the transcription factor can bind to, a grand canonical reservoir is set up, for which the occupational fraction can be calculated from Eq (3). This can be non-specific DNA (ns) or a reservoir for each type of site competing for that transcription factor (c). The fugacity $\lambda_i$ follows from the mass balance equation

$$P_i = N\theta_i + N^{(ns)}\theta_i^{(ns)} + N^{(c)}\theta_i^{(c)}(+\cdots) \tag{4}$$

where the fugacity needs to be chosen such that the expected number of transcription factors $i$ over the different reservoirs is self-consistent with $P_i$, the copy number of that protein in the cell. This equation is a polynomial in $\lambda_i$ and can be solved algebraically, when the number of reservoirs is limited, or numerically. Mathematically, the chemical potential acts as a Lagrange multiplier for the constraint given by conservation of mass. The fugacity of the regulatory transcription factor therefore depends on the number and binding energy of its specific binding sites, of any competitive genes or molecules that also bind the transcription factor, and to the reservoir of non-specific binding sites on the DNA. Transcription factors such as LacI tend to be poorly soluble in water [67], and therefore we assume that the fraction of transcription factors unbound from the DNA is negligible. It is therefore convenient to set the reference state of the free energy equal to the effective binding energy of the non-specific sites [68]. It is possible to explicitly take into account the fraction of unbound transcription factors in the solution state by including a solution state reservoir in the mass balance equation.

The fraction $\theta_1(\lambda_1, \lambda_2, \ldots)/\theta_1(\lambda_1, 0, \ldots)$ has an important physical interpretation. We reiterate here that the index number 1 is reserved for RNAP, and the numbers 2 and up correspond to the transcription factors. As long as the assumption is justified that promoter activity is proportional to the RNAP occupational fraction $\theta_1$, this fraction is equal to the macroscopic activity of the gene, relative to its activity in the absence of transcription factors. The *a priori* unknown proportionality constant cancels out, and the resulting relative quantity, usually called 'fold-change', can be measured directly, as was done in, e.g., Brewster *et al.* [28]. To highlight its importance, we introduce the fold-change $\Phi$ as the fraction $\theta_1(\lambda_1, \lambda_2, \ldots)/\theta_1(\lambda_1, 0, \ldots)$ explicitly.

In Fig 1B we show the possible states and their statistical weights in the grand canonical ensemble for a gene regulated by the simplest nontrivial regulation architecture, refered to as 'simple repression'. Such an architecture consists of a promoter sequence, partially overlapping with an operator site for a repressor. RNAP can bind to the promoter with binding energy $\epsilon_1$, and a repressor can bind to the operator site with energy $\epsilon_2$, while the simultaneous binding of both RNAP and repressor is prohibited by excluded volume interactions. In this architecture,

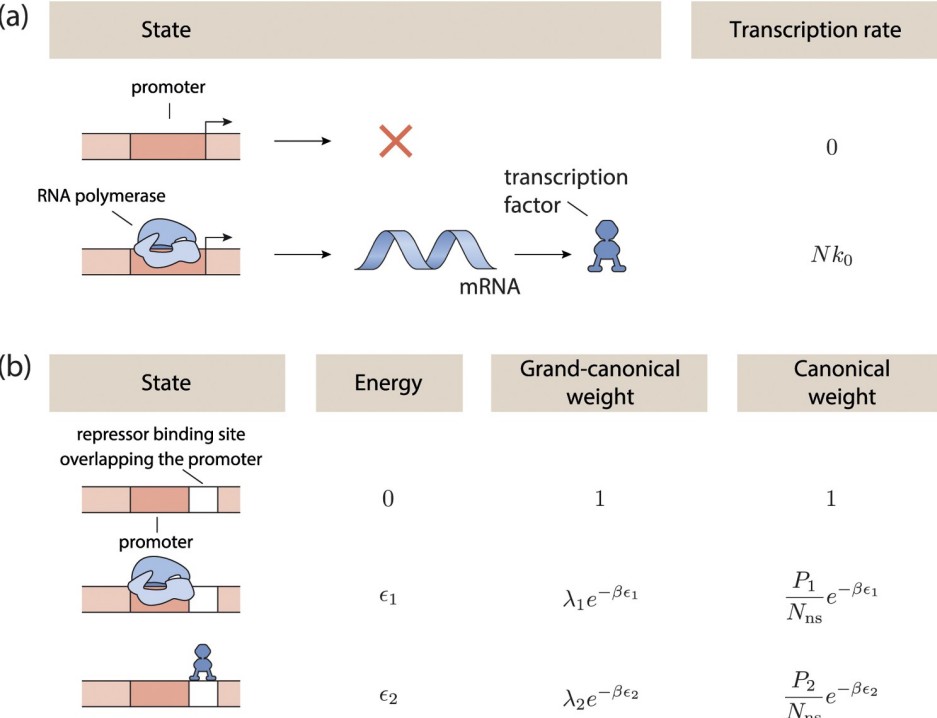

**Fig 1. Gene regulation in the grand canonical ensemble. a** Transcription rate is assumed to be dependent on the configurational state of the genome. When RNA polymerase (RNAP) is bound to the promoter sequence, transcription is initiated with a constant rate equal to the product of $k_0$, the (constant) transcription rate of an individual RNAP molecule and $N$ the total number of genes encoding for this protein. When RNAP is not bound to the promoter sequence, no transcription is initiated. **b** States and weights diagram of a gene regulated by 'simple repression'. Three configurational states are possible in this architecture, with the free energy of this configuration and the statistical weight of this configuration given, both in the canonical and grand-canonical ensembles. The configurational state where both RNAP (1) and repressor (2) are bound is forbidden because of excluded volume interactions.

the grand canonical partition function is given by

$$\Xi = 1 + \lambda_1 e^{-\beta \epsilon_1} + \lambda_2 e^{-\beta \epsilon_2}, \tag{5}$$

with $\epsilon_{1,2}$ the binding (free) energies of RNA polymerase and the repressor respectively.

In the case of the gene regulated by 'simple repression', the occupational fraction of RNA polymerase, in the presence and absence of the regulatory transcription factor $i$, are respectively given by

$$\theta_1(\lambda_1, \lambda_2) = \frac{\lambda_1 e^{-\beta \epsilon_1}}{1 + \lambda_1 e^{-\beta \epsilon_1} + \lambda_2 e^{-\beta \epsilon_2}}, \tag{6}$$

$$\theta_1(\lambda_1, 0) = \frac{\lambda_1 e^{-\beta \epsilon_1}}{1 + \lambda_1 e^{-\beta \epsilon_1}}. \tag{7}$$

The fold-change function of this gene, $\Phi = \Phi(\lambda_1, \lambda_2)$ is then given by the ratio of the RNAP occupational fraction in the presence and absence of the regulating transcription factor,

$$\Phi = \frac{\theta_1(\lambda_1, \lambda_2)}{\theta_1(\lambda_1, 0)} = \frac{1 + \lambda_1 e^{-\beta\epsilon_1}}{1 + \lambda_1 e^{-\beta\epsilon_1} + \lambda_2 e^{-\beta\epsilon_2}} \tag{8}$$

$$\simeq \frac{1}{1 + \lambda_2 e^{-\beta\epsilon_2}}, \qquad (\lambda_1 e^{-\beta\epsilon_1} \ll 1) \tag{9}$$

The binding free energy of RNA polymerase to its specific site is relatively weak compared to its binding at non-specific sites on the DNA [69], and as such, the situation where $\lambda_1 \exp{-\beta\epsilon_1} \ll 1$, is rather general. This allows us to make the final simplification, known as the 'weak promoter limit', in the fold-change expression. Because of this simplification, we can in most circumstances keep the fugacity of RNA polymerase unspecified, as it does not influence the relative activity of the gene. As such, RNA polymerase can usually be left out of the circuit.

In Section B in S1 Appendix we have provided step-by-step worked examples on how to calculate the fold-change function $\Phi$ and the transcription factor fugacities $\lambda_i$ for a number of relevant regulatory architectures.

## Rate equations

The change in the copy number of a protein P is given by two contributions. The first contribution stems from the first order degradation of the protein by the cellular recycling machinery. The second is the synthesis of P by the ribosomes, the rate of which is proportional to the concentration of P-encoding mRNA (M). The rate equation describing the copy numbers **P** of a set of proteins involved in a genetic circuit is given by

$$\frac{d\mathbf{P}(t)}{dt} = -\Gamma_P \mathbf{P}(t) + k_r \mathbf{M}(t), \tag{10}$$

with **P**, **M** the copy numbers of protein and mRNA respectively. Furthermore, $\Gamma_P$ and $k_r$ are diagonal matrices containing the first order degradation rate constants, and the activity rate constants of the ribosomes. The mRNA is produced by RNA polymerase at the gene of interest at a rate that depends on the arrangement of transcription factors. For all proteins involved in the genetic circuit of interest, these rates are given by the vector **k**. Since transcription—the synthesis of mRNA—and the transport to the ribosomes takes a finite amount of time, we explicitly introduce a time delay $\tau$ to account for that delay caused by transcription and transport. At the same time, mRNA also undergoes first order degradation by the cellular machinery with rate constants given by the diagonal matrix $\Gamma_M$. The mRNA degradation rate constants are typically higher in magnitude compared to the protein rate constants $\Gamma_P$. However, in many cases the dominant mechanism causing first order decay is dilution due to a global growth rate, in which case the mRNA and protein degradation rates are comparable [15]. In the latter case, active degradation mechanisms may still alter the individual degradation rate constants. The rate equation describing the change in mRNA copy numbers **M** is then given by

$$\frac{d\mathbf{M}(t)}{dt} = -\Gamma_M \mathbf{M}(t) + \mathbf{k}(\mathbf{P}(t - \tau)). \tag{11}$$

This rate equation is a delay differential equation (DDE) [42], since it depends on the transcriptional activity of the genes **k** at a time in the past. The rate constants **k** depend nonlinearly on the concentration of transcription factors that influence the transcription of the gene. We

take the individual transcription rates $k_i$ as proportional to the (equilibrium) probability that their promoter region is occupied by RNA polymerase (RNAP), an assumption which is justified when the formation of the RNAP open complex on the promoter site of a gene is slow in comparison to the binding and unbinding kinetics of transcription factors over the genome [49–52]. Using these assumptions, the rates of mRNA synthesis can be approximated by

$$k_i(t) = k_i^{(0)} N_i \theta_1^{(i)}(\mathbf{P}(t)),\tag{12}$$

where $k_i$ are the entries of the vector $\mathbf{k}$. Furthermore, $k_i^{(0)}$ is the (constant) rate at which mRNA of protein $i$ is produced when RNAP reads the promoter sequence, $N_i$ is the number of gene copies contributing to the production of the protein and $\theta_1^{(i)}$ the occupational fraction of the promoter of gene $i$ by RNA polymerase, calculated as the fraction of all configurational microstates where the RNA polymerase occupies the promoter sequence. As such, the occupational fraction $\theta_1^{(i)}$ (and by extension, $k_i$) is implicitly a function of time, since it is a function of all transcription factor copy numbers $\mathbf{P}$ affecting this binding probability at a given time.

When the gene is completely unregulated, that is, in the absence of any regulatory proteins, the concentrations of protein and mRNA will reach a steady-state. In that case, we can write for each protein $i$ the two steady-state equations

$$\begin{cases} (\Gamma_{\mathrm{P}})_{ii} P_i^{(0)} &= (k_{\mathrm{r}})_{ii} M_i^{(0)}, \\ (\Gamma_{\mathrm{M}})_{ii} M_i^{(0)} &= k_i^{(0)} N_i \theta_1^{(i)}(0), \end{cases}\tag{13}$$

Here, $M_i^{(0)}, P_i^{(0)}$ are the steady-state unregulated copy numbers of mRNA and the protein $i$ it encodes respectively, in the absence of any regulation. Furthermore, $(\Gamma_{\mathrm{M,P}})_{ii}$ is the $i$th diagonal element of matrix $\Gamma_{\mathrm{M,P}}$. From the steady-state equations, we can directly obtain a measure for $(k_{\mathrm{r}})_{ii}$ in terms of the unregulated steady-state copy numbers of protein and mRNA. Moreover, inserting Eq (13) into Eq (11), we obtain for the instantaneous (not steady-state) copy number of each mRNA species $i$

$$\frac{\mathrm{d}M_i}{\mathrm{d}t} = -(\Gamma_{\mathrm{M}})_{ii} M_i + (\Gamma_{\mathrm{M}})_{ii} M_i^{(0)} \frac{\theta_1^{(i)}(\mathbf{P}(t-\tau))}{\theta_1^{(i)}(0)}.\tag{14}$$

Here we recognise the fraction $\theta_1^{(i)}(\mathbf{P})/\theta_1^{(i)}(0)$ as the fold-change fraction $\Phi$ from the previous section. We generalise $\Phi$ to the fold-change vector $\mathbf{\Phi}$, defined by the set of fractions $\theta_1^{(i)}(\mathbf{P})/\theta_1^{(i)}(0)$ for each protein involved in the genetic circuit. Here each fraction is acting as the input-output function of that specific gene, depending on the architecture of the specific gene promoter sequence. The fold-change vector is implicitly a function of all the protein fugacities in the system, which in turn depend on the protein concentrations $\mathbf{p}$, hence our choice to write $\mathbf{\Phi}$ explicitly as a function of $\mathbf{p}$.

It is convenient here to normalise the copy numbers of proteins and mRNA by their steady-state unregulated copy numbers. We therefore introduce the normalised copy numbers $\mathbf{p}, \mathbf{m}$ which are defined for each element in the vectors by $p_i \equiv P_i/P_i^{(0)}, m_i \equiv M_i/M_i^{(0)}$, so that

we obtain the following system of DDEs.

$$
\begin{cases}
\dfrac{d\mathbf{m}(t)}{dt} &= -\Gamma_M \mathbf{m}(t) + \Gamma_M \boldsymbol{\Phi}(\mathbf{p}(t-\tau)) \\[2mm]
\dfrac{d\mathbf{p}(t)}{dt} &= -\Gamma_P \mathbf{p}(t) + \Gamma_P \mathbf{m},
\end{cases}
\tag{15}
$$

There are three relevant cases to consider. When a stable steady-state is reached, $\mathbf{p}$ is determined only by the fold-change $\boldsymbol{\Phi}$. Alternatively, when $\Gamma_M \gg \Gamma_P$, the concentrations of mRNA reach a steady-state fast and the system of rate equations reduces to

$$
\frac{d\mathbf{p}(t)}{dt} = -\Gamma_P \mathbf{p}(t) + \Gamma_P \boldsymbol{\Phi}(\mathbf{p}(t-\tau)). \quad (\Gamma_M \gg \Gamma_P)
\tag{16}
$$

In the case that both degradation constants are comparable, the full system of Eq (15) should be used. The quantity called fold-change acts as the input-output function and replaces the Hill function used in the literature. The form of the fold-change depends on the regulatory architecture of interest. It does dependent on the copy numbers of transcription factors that are involved, and it is through this dependence that the dynamics of one gene can be coupled to a different gene.

## Results & discussion

### The Goodwin oscillator

A simple nontrivial feedback system was proposed by Goodwin *et al.* [70] and is currently seen as a general feedback model in biology [71]. A gene in this feedback system, which bears the name 'Goodwin oscillator', directly produces a transcription factor which inhibits its own transcription. The promoter architecture follows the 'simple repression' scenario, which we schematically show in Fig 2B. The fold-change function corresponding to this architecture was derived above and is given by Eq (9).

We can numerically integrate the rate equations (see Eq (15)) for different gene copy numbers using the method of steps, starting with a protein and mRNA copy number of 0 over the time interval $[-\tau, 0]$. The transcription factor binding free energy was -15 $k_B$T for the specific sites. The steady-state unregulated repressor copy number $P^{(0)}$ was 5 per gene copy. For these trajectories, we did not include any other competitor genes. Finally, the protein and mRNA degradation rates $(\Gamma_P)_{ii}$, $(\Gamma_M)_{ii}$ were 0.03 min$^{-1}$ and the time delay $\tau$ was 18.5 min as per ref [15]. The resulting trajectories are shown in Fig 2.

In Fig 2 we see that for a low copy number of genes, the concentrations of mRNA and repressor quickly proceed to a stable steady state, with an initial overshoot that is corrected for quickly. This can graphically be seen in the phase space figure in Fig 2C. The phase space trajectory quickly spirals to the stable point at the intersection of the nullclines. For higher gene copy numbers, we see first the same qualitative picture, although the oscillations dampen out at a lower rate. Above a certain threshold copy number, the oscillations become self-sustained and we see the phase space trajectory approach a stable limit cycle, centred around the intersection between the nullclines. These self-sustained oscillations only appear with a nonzero time delay $\tau$.

Self-sustained oscillations can also be attained with just a single gene copy, in the presence of a reservoir of competitor genes. In this case we numerically integrate Eq (15) for a single gene copy in the presence of a number of competitor sites. We used a binding energy to the competitor sites of -18 $k_B$T, and have set the steady-state unregulated repressor copy number

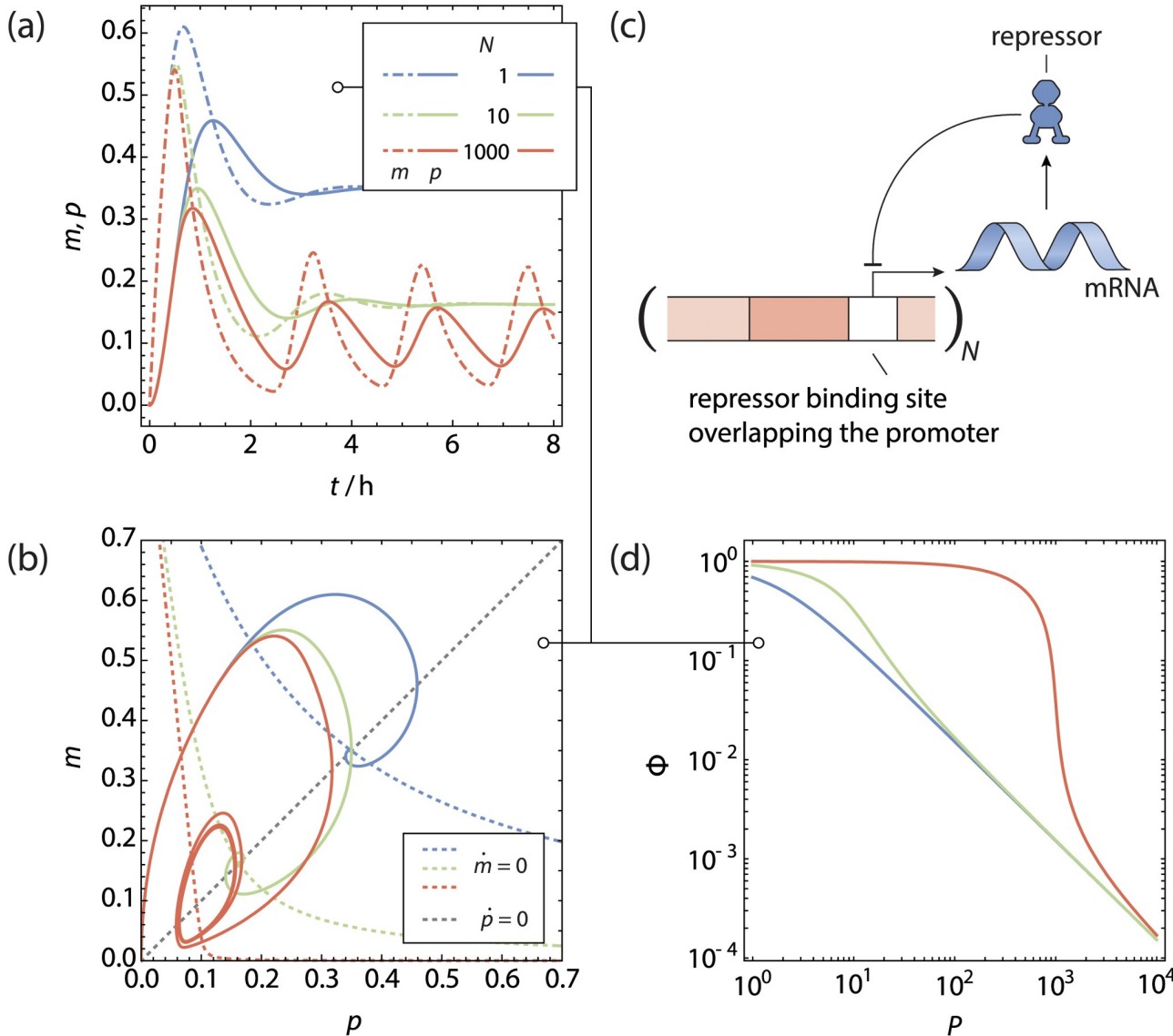

**Fig 2. The copy number titrating oscillator. a** Normalised Protein (solid lines) and mRNA (dotdashed lines) concentration as function of time for a gene regulated by **b** simple repression scenario with $N$ gene copies, producing its own repressor. **c** Phase space trajectories of **a**, shown in conjunction with the nullclines. Because of the time delay in evaluating the magnitude of $\dot{m}$, the phase space trajectories do not cross the $\dot{m}$-nullcline completely horizontally. For a sufficiently high gene copy number a stable limit cycle is reached. **d** Fold-change of the gene as function of the total number of transcription factors.

$P^{(0)}$ to 100 per gene copy. A high baseline activity is necessary here: without sufficient repressors in the cell, the gene will be almost completely outcompeted by the competitor sites. All other assumptions were kept fixed as before. The resulting trajectory are plotted in Fig 3. The results are similar to the previous case: when there are multiple binding sites competing for the repressor, the response curve becomes steeper and self-sustained oscillations are attained.

Finally, the promoter architecture itself can also generate sufficient nonlinearity to sustain oscillations in a genetic feedback loop that only involves a single gene copy. For example, in the looping architecture [72], transcription factors have two DNA binding domains that are

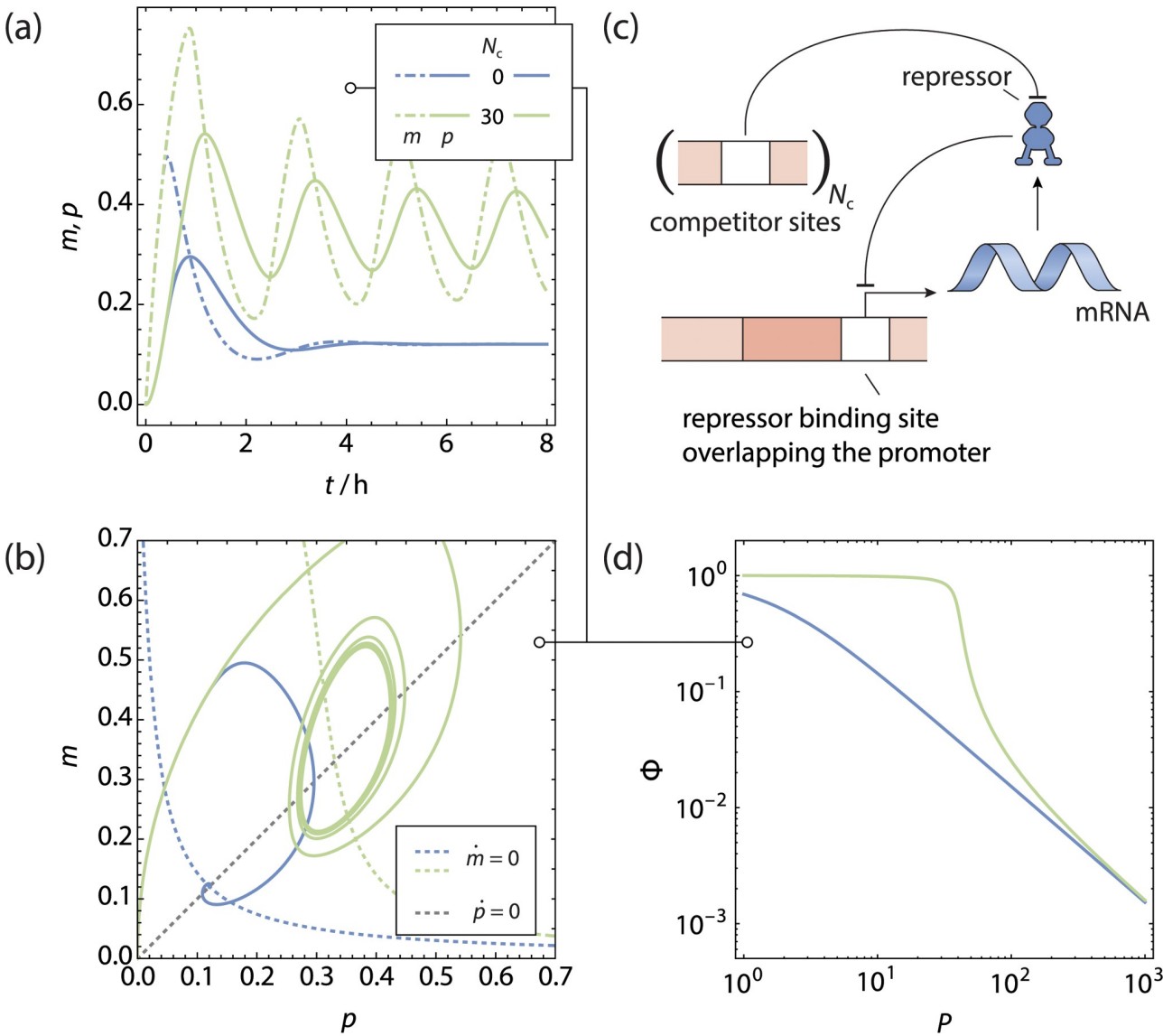

**Fig 3. The single gene oscillator. a** Normalised protein (solid lines) and mRNA (dotdashed lines) concentration as function of time for a gene regulated by **b** simple repression scenario producing its own repressor, with $N_c$ sites competing for a common pool of repressors. **c** Phase space trajectories of **a**, shown in conjunction with the nullclines. Because of the time delay in evaluating the magnitude of $\dot{m}$, the phase space trajectories do not cross the $m$-nullcline completely horizontally. For a sufficiently high competitor copy number a stable limit cycle is reached. **d** Fold-change of the gene as function of the total number of transcription factors. Note that for the qualitative behaviour, it does not matter whether the competitor sites are DNA binding sites, enzymes or other ligands binding the repressors.

capable of binding two operator sites simultaneously. An auxiliary operator site near the promoter can enhance the efficacy of the transcription factor by increasing the probability of occupancy of the main operator site, where it is able to regulate transcription, by allowing for loops in the DNA between the operator sites. In our previous work [30], we showed that the equation describing the fold-change in the grand canonical ensemble is given by

$$\Phi = \frac{1 + \lambda x^{(a)}}{1 + \lambda(x^{(a)} + x^{(m)} + x^{(a)}x^{(a)}x_L) + \lambda^2 x^{(a)}x^{(m)}}. \tag{17}$$

Here, $x^{(m,a)} = \exp(-\beta\epsilon^{(m,a)})$ are the Boltzmann factors of the binding free energies of the transcription factor to the main (m) and auxiliary (a) operator sites. Furthermore, $x_L = \exp{-\beta\Delta F_L}$ is the Boltzmann factor of the free energy of forming a transcription factor-DNA loop spanning the auxiliary and main operator sites. $\Delta F_L$ therefore reflects a weighted average of a number of conformational states where a single transcription factor binds the two operator sites simultaneously. As before, we have the transcription factor fugacity, $\lambda = \exp\beta\mu$ that is found self-consistently by applying mass conservation.

Forming a DNA loop between two bound sites on a transcription factor causes the DNA to adopt an entropically unfavourable conformation. The free energy penalty $\Delta F_L$ associated with this loop depends on the length of the DNA included within the loop [73]. We numerically integrate the rate Eq (15) in Fig 4 for different values of $\Delta F_L$, ranging between 7 $k_B$T and 11 $k_B$T, starting with protein and mRNA copy numbers of 0. As before, the transcription factor binding free energy was -15 $k_B$T for both the main and auxiliary operator site, and the steady state unregulated repressor copy number $P^{(0)}$ was 5 per gene copy. We observe self-sustained oscillations for the two lower looping free energies, with a very similar oscillation period. When the looping free energy is too high, the statistical weight of the looped configuration lowers and a stable steady state is observed instead.

## Stability analysis

An important question is whether it is possible to predict, a priori, whether a given genetic feedback circuit will lead to self-sustained oscillations. This is of particular importance in applications, since transcription is inherently stochastic and especially fluctuations in the time delay lead to irregular availability of mRNA. The traditional approach to evaluate the stability of stationary points in systems of ordinary differential equations (ODEs) [74] can be extended to DDEs, following the approach of Hale *et al.* [42]. In the context of genetic circuits, we will follow this approach to analyse a generalised model for $n$ genes that interact through a common set of transcription factors. We will explicitly allow fluctuations in the time delays. In the system of equations from Eq (15), we now explicitly introduce time delays both in the transcription ($\tau_P$) as well as in the translation ($\tau_M$) steps. The model then becomes

$$\begin{cases} \dot{\mathbf{m}}(t) & = -\Gamma_M\mathbf{m}(t) + \Gamma_M\,\Phi(\mathbf{p}(t-\tau_P)) \\ \dot{\mathbf{p}}(t) & = -\Gamma_P\mathbf{p}(t) + \Gamma_P\mathbf{m}(t-\tau_M), \end{cases} \tag{18}$$

where as before, $\mathbf{m}$, $\mathbf{p}$ are vectors built up from $m_i$, $p_i$, and $\Gamma_M$, $\Gamma_P$ are diagonal matrices containing the decay rate constants for each protein and mRNA type. Here, by abuse of notation, we consider the fold-change $\Phi$ implicitly as function of all proteins in $\mathbf{p}$. The model could be generalised to include independent time delays for all mRNA and protein species $i$ in a straightforward way. The fold-change for each gene depends on the architecture, and in principle on all proteins in $\mathbf{p}$. Suppose that we found a stationary point at the intersection of the nullclines at $(\mathbf{m}^*, \mathbf{p}^*)$. In order to investigate the stability of this stationary point, we will linearise the differential equations around this point and find special trajectories in the neighbourhood of the stationary point of the form $\mathbf{v}e^{\lambda t}$. These trajectories move exponentially outward or inward to the stationary point, with a direction $\mathbf{v}$ which is to be determined later. We linearise the fold-change term in the equation for $\dot{\mathbf{m}}$ around $(\mathbf{m}^*, \mathbf{p}^*)$, which gives us the Jacobian with $n \times n$ matrix with elements given by

$$J_{ij} = (\Gamma_M)_{ii}\frac{\partial\Phi_i(\mathbf{p})}{\partial p_j}, \tag{19}$$

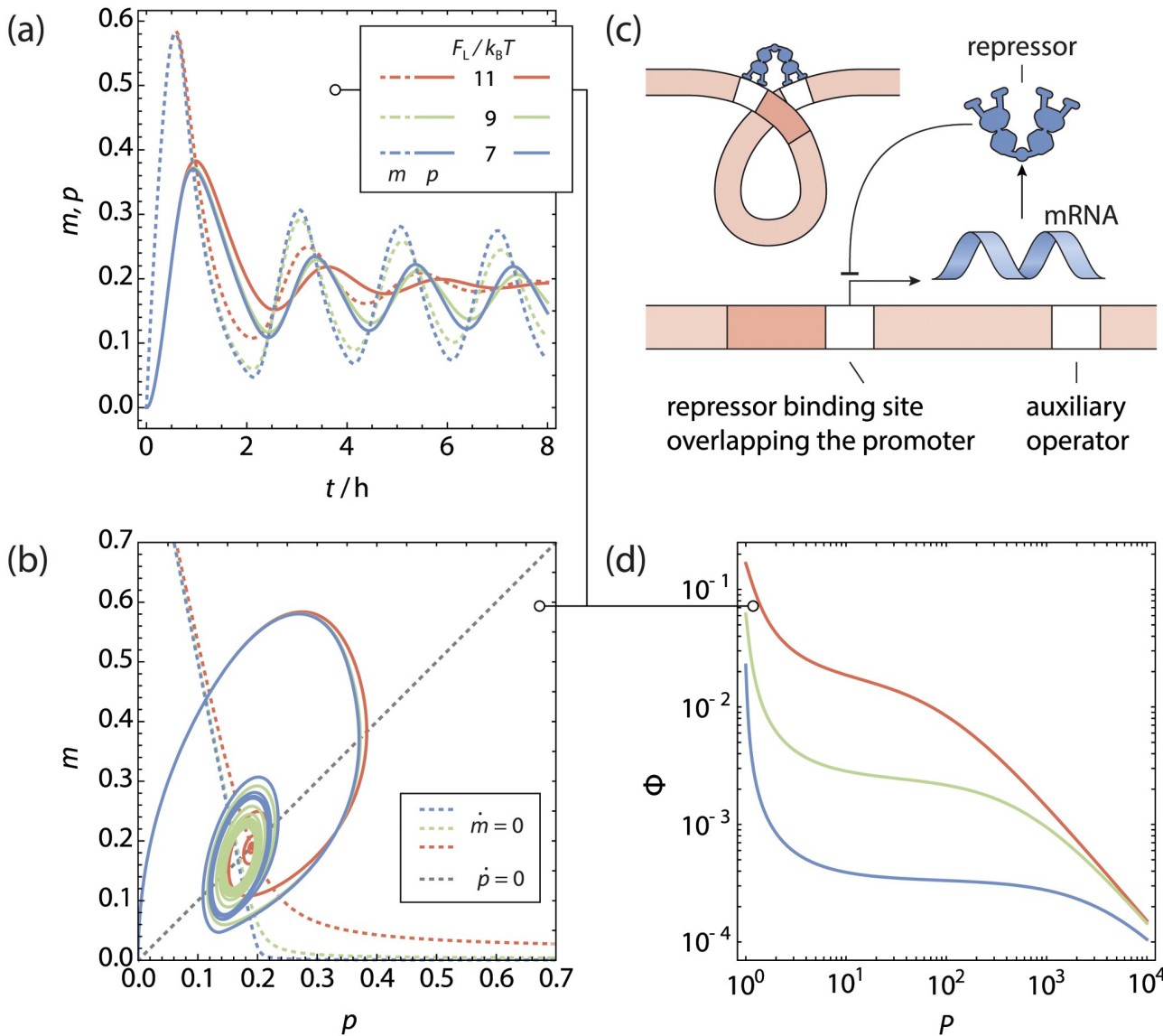

**Fig 4. The looping gene oscillator. a** Normalised protein and mRNA concentration as function of time for a gene regulated by **b** a looping scenario producing its own repressor. **c** Phase space trajectories of **a**, shown in conjunction with the nullclines. Because of the time delay in evaluating the magnitude of $\dot{m}$, the phase space trajectories do not cross the $\dot{m}$-nullcline completely horizontally. For a sufficiently favourable looping free energy a stable limit cycle is reached. Here, the steady-state unregulated repressor copy number was kept at $P^{(0)} = 5$. **d** Fold-change of the gene as function of the total number of transcription factors.

evaluated at $(\mathbf{m}^*, \mathbf{p}^*)$. To find trajectories in the neighbourhood of the stationary point that behave as $\mathbf{v}e^{\lambda t}$, we make the substitutions $\mathbf{m}(t) = \mathbf{a}e^{\lambda t}$, $\mathbf{p} = \mathbf{b}e^{\lambda t}$ in the linearised equation, with $\lambda$ a complex scalar. This allows us to write the linearisation of Eq (18) around $(\mathbf{m}^*, \mathbf{p}^*)$ in the form of a linear matrix equation

$$\lambda \begin{pmatrix} \mathbf{a} \\ \mathbf{b} \end{pmatrix} e^{\lambda t} = \underbrace{\begin{pmatrix} -\Gamma_M & Je^{-\lambda \tau_P} \\ \Gamma_P e^{-\lambda \tau_M} & -\Gamma_P \end{pmatrix}}_{\mathbf{A}} \begin{pmatrix} \mathbf{a} \\ \mathbf{b} \end{pmatrix} e^{\lambda t} \tag{20}$$

The factors $e^{\lambda t}$ cancel out, and it remains to find the eigenvalues $\lambda$ of the $2n \times 2n$ matrix $\mathbf{A}$. The eigenvalues can be found by solving the characteristic equation $\det \mathbf{A} - \lambda \mathbf{I} = 0$, with $\mathbf{I}$ the identity matrix.

$$\det \begin{pmatrix} -\Gamma_{\mathrm{M}} - \lambda \mathbf{I} & Je^{-\lambda \tau_{\mathrm{P}}} \\ \Gamma_{\mathrm{P}} e^{-\lambda \tau_{\mathrm{M}}} & -\Gamma_{\mathrm{P}} - \lambda \mathbf{I} \end{pmatrix} = 0. \tag{21}$$

The roots of this equation in the complex plane determine the stability of the point ($\mathbf{m}^*$, $\mathbf{p}^*$). If all the roots have negative real parts, then all exponential trajectories around the stationary point move inward and the stationary point is asymptotically stable. There are finitely many dominant roots, i.e., roots with the same real part such that the real parts of all other roots are strictly less than the real part of the dominant roots. The stability contour can be traced by logging when dominant roots cross the imaginary axis in the complex plane. If a dominant root crosses the imaginary axis with nonzero imaginary part, i.e., the real part of a complex dominant root crosses 0, a Hopf bifurcation usually occurs and oscillatory behaviour is seen.

For a circuit with a single gene in a negative feedback loop, Eq (21) simplifies to

$$\frac{(\lambda + \Gamma_{\mathrm{M}})(\lambda + \Gamma_{\mathrm{P}})}{\Gamma_{\mathrm{M}} \Gamma_{\mathrm{P}}} e^{\lambda \tau} - \frac{d\Phi(p^*)}{dp} = 0, \tag{22}$$

with $\tau = \tau_{\mathrm{M}} + \tau_{\mathrm{P}}$ the total time delay, a confirmation that only the total time delay in the feedback loop determines the local stability. Here we have abused the notation somewhat and let $\Gamma_{\mathrm{M,P}}$ denote the scalar value of the single element each matrix contains in this single-membered circuit. It will turn out useful to make the equation dimensionless by scaling time by the factor $\gamma \equiv \sqrt{\Gamma_{\mathrm{M}} \Gamma_{\mathrm{P}}}$ and redefining $\lambda$ as the rescaled eigenvalue, resulting in the equation

$$\left(\lambda + \frac{\Gamma_{\mathrm{M}}}{\gamma}\right)\left(\lambda + \frac{\Gamma_{\mathrm{P}}}{\gamma}\right) e^{\lambda \gamma \tau} - \frac{d\Phi(p^*)}{dp} = 0. \tag{23}$$

In Fig 5 we plot the regions where self-sustained oscillations are possible and where the stationary point leads to a stable steady-state. The line along which the real part of the dominant eigenvalues $\lambda$ equal 0—separating the two regions—is referred to as the stability contour, and was calculated by numerically solving Eq (23). The stability contour depends only on the rescaled time delay $\gamma \tau$ and the slope of the fold-change in the stationary point. We have taken $\Gamma_{\mathrm{M}} = \Gamma_{\mathrm{P}} = \gamma$ in this graph for convenience, but the figure does not significantly change when the two degradation constants are of comparable magnitude. When the delay is comparatively small, stable oscillations can only occur if the slope of the fold-change with respect to the normalised transcription factor copy number is very steep, corresponding to strong cooperativity or competition. At much larger delays, this requirement is less strict, although the slope should be steeper than −1. As a consequence, a gene without a cooperative architecture will be unable to sustain oscillatory behaviour in isolation. (The slope of the fold-change curve for a gene with an effective Hill exponent of 1 never exceeds −1 with respect to the normalised repressor copy number).

Close around the stationary point, the system oscillates with a period of $2\pi/\mathrm{Im}\{\lambda\}$. We show the positive branch of the real part of $\lambda$ in Fig 5A. However, since there are finitely many dominant roots, stable limit cycles do not oscillate infinitesimally close around the stationary point in phase space. Due to the Hopf bifurcation theorem, the oscillation occurs around the points in phase space where the real part of the dominant eigenvalue equals 0, that is, where the

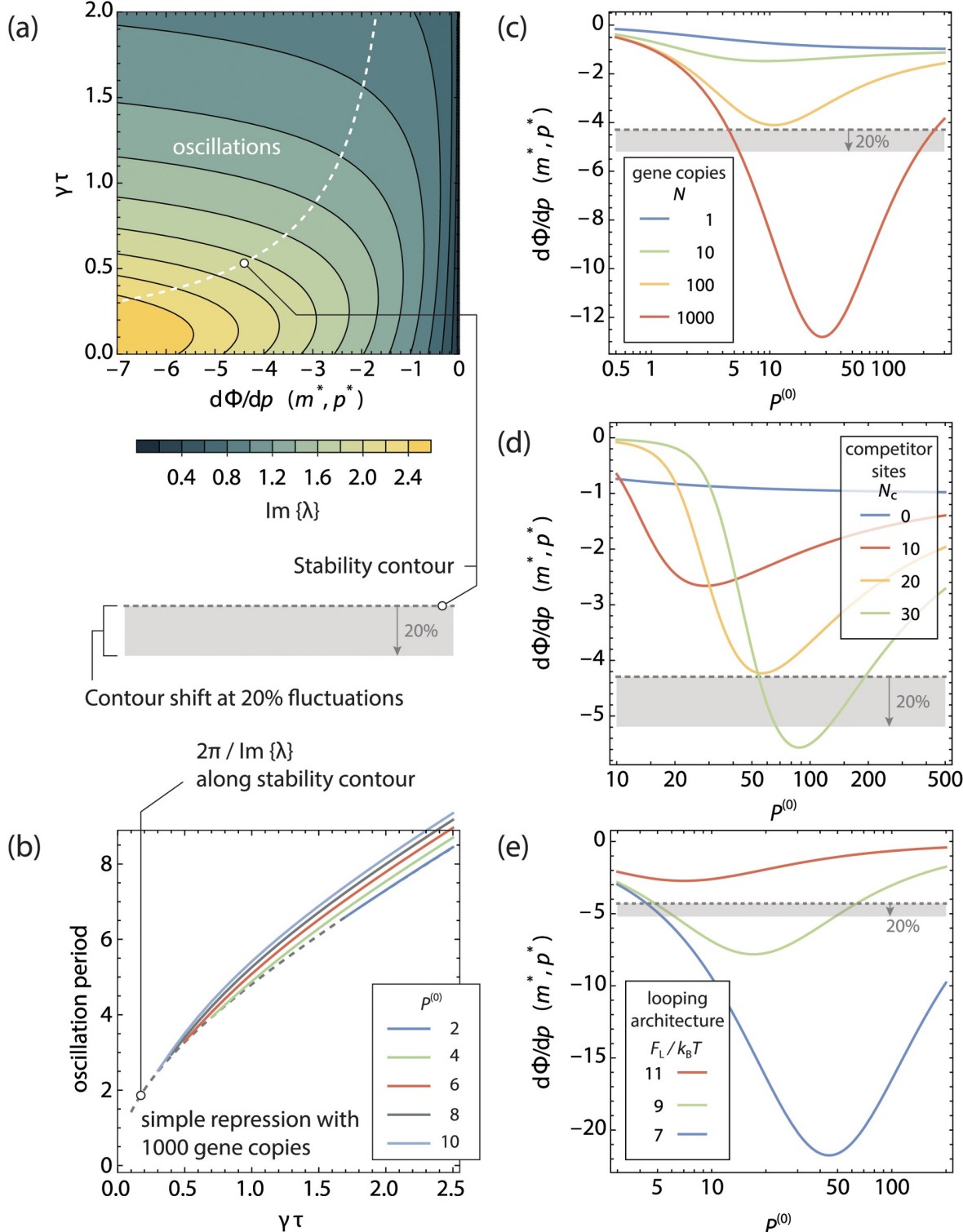

**Fig 5. Stability contours for a single gene oscillator. a** A map of stable oscillations in the phase space of normalised delay time $\gamma\tau$ versus the slope of the fold-change in the stationary point of the system $\mathrm{d}\Phi/\mathrm{d}p(m^*, p^*)$. Stable oscillations are expected when the dominant roots of Eq (23) have a positive real part. The dotted line gives the contour where the real part of the dominant roots of Eq (23) is 0. The contours indicate the magnitude of the imaginary part of the dominant root $\lambda$, which is related to the oscillation period. **b** From the Hopf bifurcation theorem, it follows that in first order approximation, the period of the oscillation is given by $2\pi$ divided by the imaginary part of the dominant roots $\lambda$ at the stability contour, shown with the dashed line. The solid lines denote the period of a number of numerical

integrations for different $P^{(0)}$. The y-axis is given in units of dimensionless time $\gamma t$. **c,d,e** Slope of the fold-change in the stationary point for **c** $N$ copies, **d** a single copy in the presence of $N_c$ competitor sites and **e** a single copy with a Looping architecture in the absence of competitor sites, of a gene regulated by its own product. Only when the slope in the stationary point crosses the threshold given by the stability contour in **a** are self-sustained oscillations possible. The grey region denotes the extent the stability contour shifts when a 20% variation is present in $\gamma\tau$.

oscillation does not grow or contract. As such, a better prediction of the period of a self-sustained oscillation is given by the $2\pi$ over the imaginary part of the dominant eigenvalue along the stability contour, as we plot in Fig 5B, together with the oscillation period obtained from numerical integrations of Eq (15). We see that indeed the oscillation period is always equal to or slightly larger than $2\pi/\text{Im}\{\lambda\}$.

Fluctuations in $\gamma\tau$ temporarily shift the stability contour locally: a system with a slope in the stationary point on the oscillation side of the stability contour could cross the boundary and become locally attractive. The oscillation of the system is expected to be erratic in such cases and lose its periodicity. On average, the stationary point is unstable, and the system tends away from it, but on short timescales the system may collapse back on the stationary point. The magnitude of the fluctuations in $\gamma\tau$ translate to a range of slopes for which this erratic behaviour is expected. In Fig 5C, 5D and 5E we have plotted the slope in the stationary point of the copy number titrating oscillator (C), the single gene oscillator (D) and the looping oscillator (E), as a function of the steady state unregulated copy number of repressors per gene copy $P^{(0)}$. The grey dotted line denotes the stability contour for $\gamma\tau = 0.555$ as per ref [15], with the grey region below denoting the shift in the stability contour upon a local 20% fluctuation in $\gamma\tau$ as a guess for the amount of fluctuations present *in vivo*. This shift was calculated by simply taking the position of the stability contour at $0.8\gamma\tau$, and does not reflect a direct integration of stochastic differential equations. The representation in Fig 5C, 5D and 5E can show if self-sustained oscillations are possible for a given architecture, and what tolerance the system has towards variances in $\gamma\tau$ and $P^{(0)}$—varying $P^{(0)}$ shifts the location of the stationary point, and thereby the slope of the fold-change curve in the stationary point. We can see in (C) that for the conditions in Fig 2, the system only barely crosses the stability contour for $N = 1000$ and would show erratic behaviour when fluctuations of 20% are present. However, for higher $P^{(0)}$, a very wide range of self-sustained oscillations are possible for $N = 1000$. For gene copy numbers of $N \leq 100$ the slope in the stationary point does not cross the stability contour for any $P^{(0)}$, and stable steady states are expected. Similarly, in (D), we see that a single gene in the presence of 30 competitors can show self-sustained oscillations over a range of different $P^{(0)}$, while 20 competitors do not generate sufficient nonlinearity to cross the stability contour for any $P^{(0)}$. We note that the exact threshold will depend on mechanistic details such as the binding free energies of the gene and its competitor sites.

While it is perhaps less intuitive that multiple copies of the same gene can lead to a titration effect similar to the situation with competitor sites, mechanistically the titration effect in both cases is the same: the existence of one or more specific sites depletes the reservoir of free transcription factors. A reservoir of $N$ gene copies lead to the same titration curve as $N - 1$ sites competing with a single gene copy—provided the binding free energy of the competing sites is the same. However, there is one subtle difference: the additional production terms arising from the multiple gene copies shift the stationary point of the system to a different point along the titration curve. As such, one can find the optimal conditions for self-sustained oscillations at a different $P^{(0)}$ per gene.

## An experimental titration oscillator

In the work of Stricker *et al.* [47], an experimental example is presented of a genetic feedback loop that can be classified as a Goodwin oscillator. While the main results in that work detail a genetic circuit with both a positive and a negative feedback loop, oscillations are also observed in a circuit without a positive feedback loop. Here, a gene is expressed in *Escherichia coli* that produces the repressive transcription factor LacI that inhibits readout of the gene. Simultaneously, LacI also represses the production of a fluorescent reporter gene yemGFP. Under normal conditions, this circuit does not show oscillatory behaviour, but when induced with certain concentrations of IPTG, the circuit shows self-sustained oscillations.

The inducer IPTG plays an important role in the behaviour of this genetic circuit. IPTG binds competitively to LacI to prevent it from binding its specific binding site at the gene promoter [75, 76] and induces a redistribution of LacI towards the non-specific DNA [77]. When present in saturating quantities (2 mM in the medium), LacI binding to the promoter site is completely inhibited, and production of the fluorescent reporter can proceed to its unregulated steady state.

As such, the negative feedback only genetic circuit from Stricker *et al.* [47] corresponds to the scenario described as the Goodwin oscillator, where IPTG takes the place of a competitor site. We can use the same mathematical modelling as for a Goodwin oscillator regulated by a 'simple repression' architecture. The only difference is that LacI bound to IPTG still occupies a site on the non-specific DNA, and very slightly decreases the effective size of that reservoir. Provided that the number of DNA basepairs far exceeds the number of LacI repressors, we can justify neglecting this effect, and the same equations hold.

In Fig 6 we show a few numerically integrated traces of Eq (15) for this genetic circuit. In this experiment, the global growth rate of the cells is not limiting the lifetime of mRNA and protein, and as such, mRNA and protein degradation constants cannot be assumed equal. In Table A in S1 Appendix we list protein and mRNA degradation constants and IPTG and LacI binding constants taken from literature, as well as an estimate for the time delay $\tau$ from the reported oscillation period in ref [47]. That leaves the unregulated steady state production of LacI $P^{(0)}$, and the internal IPTG concentration as free parameters. In Fig 6 we have chosen as an example $P^{(0)} = 10^4$ as the lowest order of magnitude where self-sustained oscillations are observed.

In the numerically integrated traces, we observe that mRNA concentrations oscillate wildly while the concentration of LacI remains close to a steady baseline of about 10% of its unregulated steady state concentration. This is an effect of the difference in protein and mRNA degradation constant, where the latter is quickly degraded to allow its concetration to adapt very quickly to changing situations. Indeed, in Stricker *et al.* [47] (S5 Fig) we see that the oscillating cells show fluorescence oscillating with a relatively narrow amplitude, and around a baseline about an order of magnitude lower than when LacI is completely induced.

We plot the stability contour for this genetic circuit in Fig 7A in the phase space of the slope of the fold-change function in the stationary point and the product $\gamma\tau$. We see that due to the relatively small time delay $\tau$ relative to the protein and mRNA degradation constants—$\gamma\tau = 0.67$—the requirement for a steep slope of the fold-change function in the stationary point is much higher than in Fig 5. This steep slope can only be achieved at a high gene baseline activity, and consequently, cells need a $P^{(0)}$ of well above $10^3$ to show self-sustained oscillations.

Only in a narrow region around specific intracellular IPTG concentrations does the slope of the fold-change function dip below the stability contour, as can be see from Fig 7B and 7C. This is reflected by the experimental results, where a relatively modest 3-fold increase (from 0.6 mM to 2 mM) of IPTG in the external medium completely quenches observed oscillations

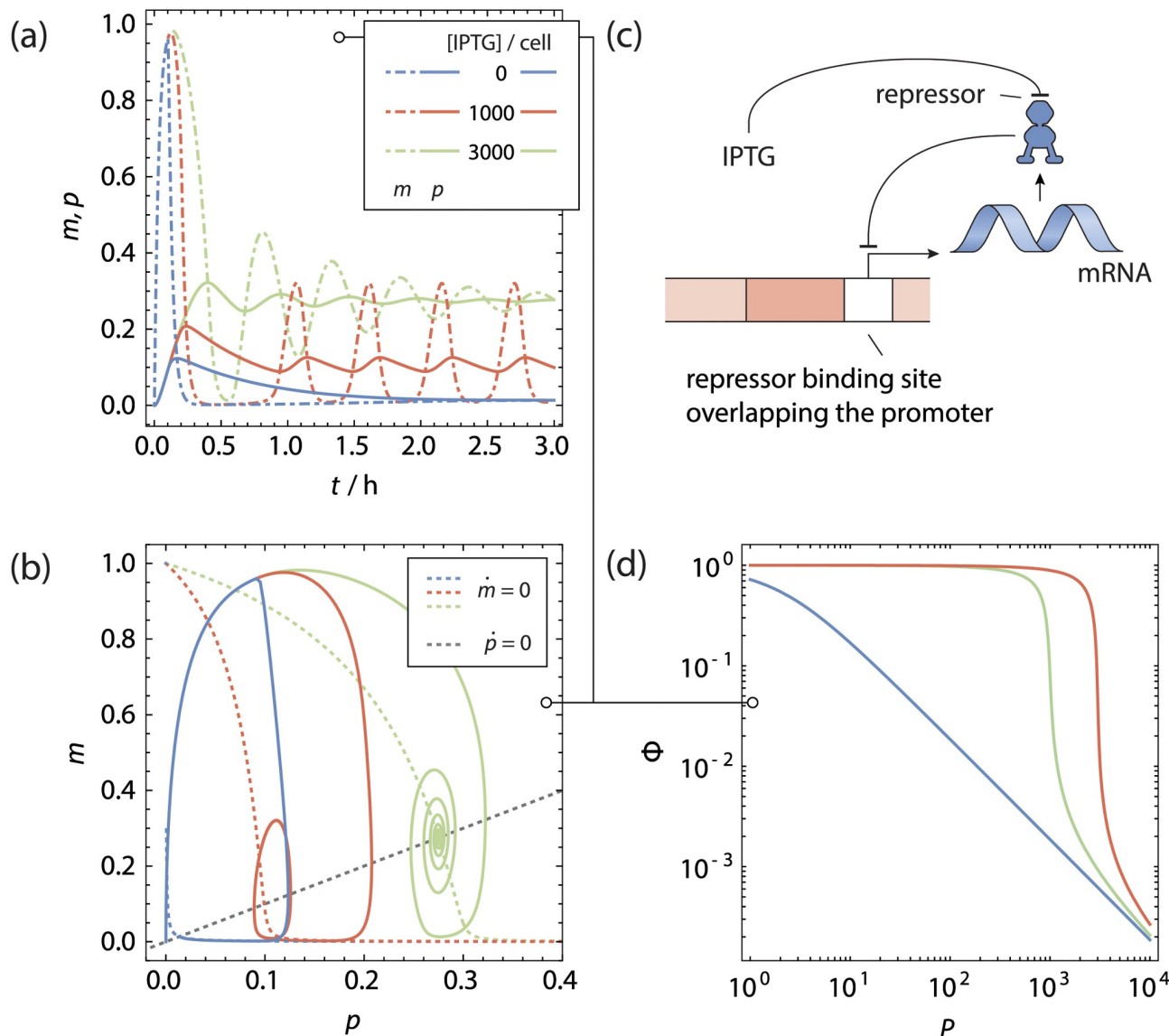

**Fig 6. Modelling the IPTG titration oscillators in Stricker et al. a** Protein (solid lines) and mRNA (dotdashed lines) copy number as function of time, and, **b** Phase space trajectories, shown in conjunction with the nullclines for a gene **c** regulated by the 'simple repression' architecture producing its own repressor. An external concentration of IPTG acts as a competing inhibitor. For a sufficiently high (internal) number of IPTG molecules in the cells a stable limit cycle is reached. **d** Fold-change of the gene as function of the total number of transcription factors.

and allows the production of fluorescent yemGFP to increase by an order of magnitude. While we do not know the absolute copy number of IPTG molecules in the cell in this experiment, it can be expected to scale linearly with the external medium IPTG concentration [78]. Indeed, a 3-fold increase of the IPTG concentration in the grand canonical model is more than enough to quench the oscillations. This is illustrated by the vertical arrow in Fig 7B: inducing the system with 1000 IPTG molecules in the cell (open circle), the stability contour is crossed and the system is expected to oscillate. Inducing with 3000 IPTG molecules (arrowhead) the stability contour is crossed once again and the system is expected to proceed to a stable steady state. Meanwhile, the fold-change $\Phi$ in the stationary point following induction increases by over an order of magnitude.

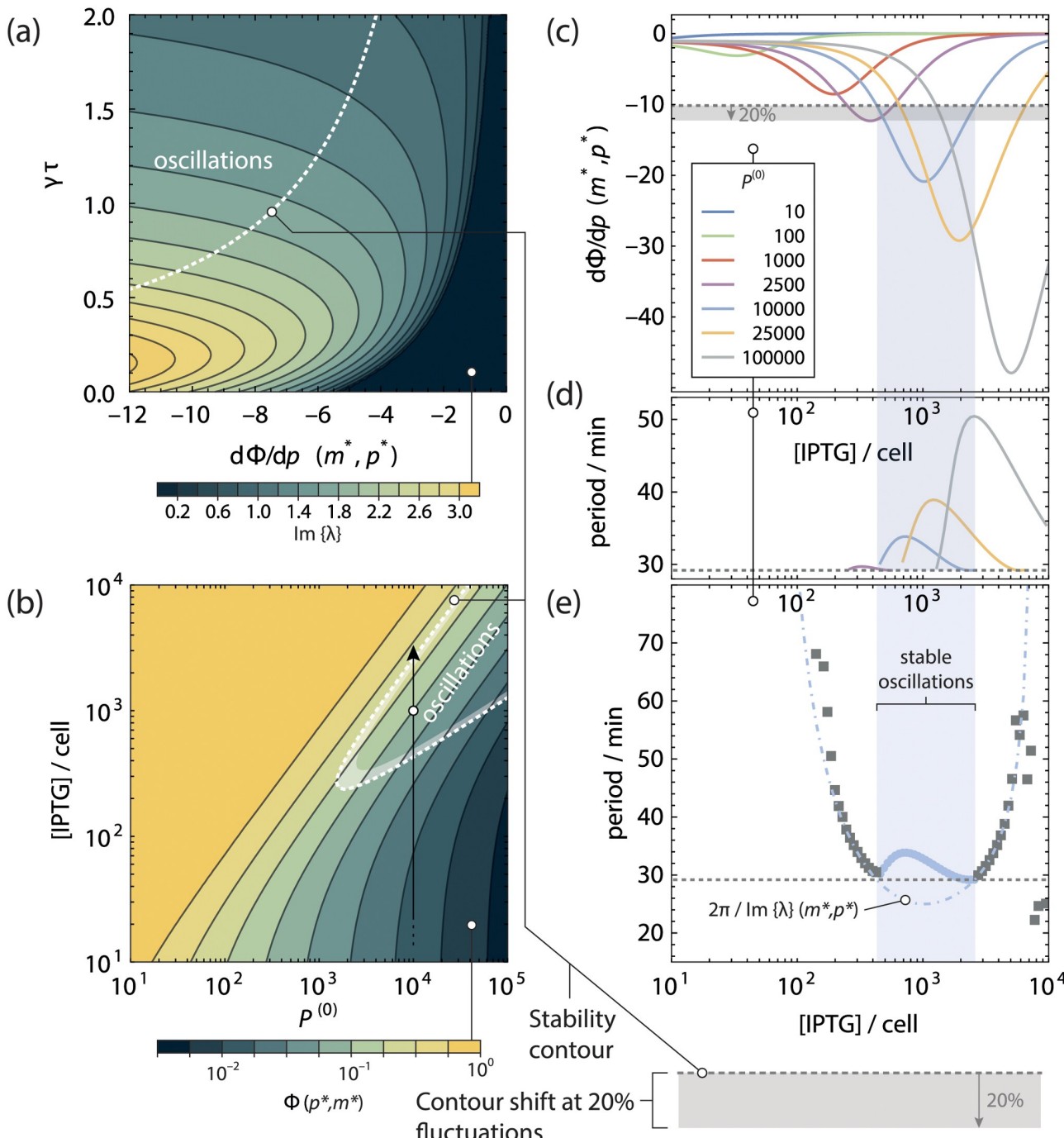

**Fig 7. Stability of the Goodwin oscillator in Stricker et al. [47]. a** Stable oscillations are expected when the dominant roots of Eq (23) have a positive real part. The dotted line gives the contour where the real part of the dominant roots of Eq (23) is 0. The colours indicate the imaginary part of the positive dominant root λ, which is related to the oscillation period. **b** Bifurcation diagram in $P^{(0)}$ and intracellular IPTG copy number. The colours denote the fold-change $\Phi(m^*, p^*)$ in the stationary point. The oscillations shown in Fig 6 lie on the solid vertical arrow: induction with $10^3$ IPTG molecules brings the system in the oscillating parameter space, while a further 3-fold increase in IPTG concentration sees the system cross the stability contour again. **c** Slope of the fold-change in the stationary point. Only when the slope in the stationary point crosses the threshold given by the stability contour in **e** are self-sustained oscillations possible. The grey region denotes the extent the stability contour shifts when a 20% variation is present in $\gamma\tau$. **d** Observed period of the oscillation as function of intracellular IPTG copy number. Close to where the slope of the fold-change crosses the stability contour, the oscillations approach $2\pi/\text{Im}\{\lambda\}$ on the stability contour (dotted line). **e** Observed period of dampened oscillations outside the stability contour for $P^{(0)} = 10^4$ (grey squares) and $2\pi/\text{Im}\{\lambda\}$ in the stationary point (light-blue dotdashed line).

While an excess of IPTG will bring the genetic circuit back to a stable steady state (see Fig 7C), the region where oscillations can be expected does become wider when the unregulated steady state activity of the gene $P^{(0)}$ is higher. Simultaneously, the slope of the fold-change in the stationary point also becomes steeper, indicating that oscillations will be more robust.

In Fig 7D and 7E we show the observed oscillation period from numerically intragrated traces of Eq (23) for this system, as a function of intracellular IPTG concentration. We see that the oscillation period approaches $2\pi/\text{Im}\{\lambda\}$ (dotted line) where the slope of the fold-change $\Phi$ in the stationary point crosses the stability contour. Outside of the stability contour, oscillations are unstable and dampen out over time, but the period of the dampened oscillations indeed neatly follows $2\pi/\text{Im}\{\lambda\}$ in the stationary point $(m^*, p^*)$ (3, grey squares). However, the oscillation period increases again when $d\Phi/dp(m^*, p^*)$ dives away from the contour. Interestingly, the extrema in oscillation period and slope do not coincide, hinting at a deeper level of complexity in their relationship.

## Conclusions

In this article we have shown that uncorrelated transcription factor binding can lead to self-sustained oscillations, even in a genetic feedback loop that only consists of a gene regulated by a simple repression architecture that directly produces its own repressor. The only condition for this to happen is that multiple transcription factor binding sites are coupled through their competition for a shared pool of transcription factors.

Here, we have implemented a mechanistic model for the fold-change of a promoter architecture, first developed in Weinert *et al.* [29] and Landman *et al.* [30]. The model allows the accurate calculation of the response of a gene to an external concentration of transcription factors in the competing limit. Importantly, the response functions follow directly from the regulatory architecture and are suitable for situations where transcription factors are shared by different promoter sites. Consequently, we no longer need to model the response of a gene with a phenomenological Hill function when the binding architecture demands a different isotherm.

We see that when a transcription factor is strongly competed for, the fold-change function of a gene that is regulated by that transcription factor becomes very sharp, even when binding is uncorrelated, that is, governed by a Langmuir isotherm. This titration effect, caused by competition, is able to provide a sufficiently steep response function for even minimal genetic feedback loops, where the gene produces its own repressor directly, to achieve self-sustained oscillations. Even though the individual gene copies are uncorrelated, they synchronise because they are coupled by a common transcription factor pool.

The main strength of our approach is the inclusion of a mechanistic gene response function within a relatively simple mathematical framework—delay differential equations—which allows us to derive a powerful stability criterion that can tell a priori whether a genetic feedback loop can show self-sustained oscillations. While DDEs do not explicitly include the fluctuations inherent in the stochastic transcription and translation processes, their use predicts the existence of an erratic regime arising from the fluctuations in transcriptional time delay. The extent by which the stability contour is crossed provides an estimation how robust the oscillations are with respect to stochastic noise in the transcription and translation process. Further, it follows from the Hopf bifurcation theorem that the stability contour also provides a first order approximation to the oscillation period of the feedback loop.

The ultimate proof that cooperativity is not required is provided by experimental observations by Stricker *et al.* [47]. Our model shows that the oscillations observed in a direct feedback

construct induced by IPTG can be interpreted as a titration oscillator stabilised by a strong competition for the transcription factor.

## Supporting information

**S1 Appendix. Transcription factor competition facilitates self-sustained oscillations in single gene genetic circuits—Supplementary materials.**
(PDF)

**S2 Appendix. Mathematica notebook used to generate all the graphs in the paper.** This file requires the use of Mathematica [79].
(NB)

## Acknowledgments

We are grateful to Aaron New for enlightening discussions and for many suggestions.

## Author Contributions

**Conceptualization:** Jasper Landman, Willem K. Kegel.

**Formal analysis:** Jasper Landman.

**Investigation:** Jasper Landman.

**Methodology:** Jasper Landman, Sjoerd M. Verduyn Lunel.

**Project administration:** Willem K. Kegel.

**Supervision:** Willem K. Kegel.

**Validation:** Jasper Landman.

**Visualization:** Jasper Landman.

**Writing – original draft:** Jasper Landman.

**Writing – review & editing:** Jasper Landman, Sjoerd M. Verduyn Lunel, Willem K. Kegel.

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
