## [Decision Letter · Decision Letter 0]

26 Jul 2023

Dear Mr. Landman,

Thank you very much for submitting your manuscript "Transcription factor competition facilitates self-sustained oscillations in single gene genetic circuits" for consideration at PLOS Computational Biology. As with all papers reviewed by the journal, your manuscript was reviewed by members of the editorial board and by several independent reviewers. The reviewers appreciated the attention to an important topic. Based on the reviews, we are likely to accept this manuscript for publication, providing that you modify the manuscript according to the review recommendations.

Please pay particular attention to the comments regarding clarity, considering the target audience. It would be beneficial to our readers if you provide a little bit more explanation about the statistical mechanics principles used here.

Sincerely,

Pedro Mendes, PhD

Section Editor

PLOS Computational Biology

Pedro Mendes

Section Editor

PLOS Computational Biology

Reviewer's Responses to Questions

**Comments to the Authors:**

Reviewer #1: The authors have worked hard to address all of my comments and suggestions on the original submission. The paper is much clearer now, although it is still heavy going. Now that I understand better what the authors are trying to establish, it is still not clear to me why they need 'grand canonical ensembles' to compute the 'fold-change' of transcription rate in the case when a TF binds to competing sites on the DNA or to a protein inhibitor. In the attached pdf, I analyze this mechanism using standard biochemical kinetic equations and the quasi-steady state approximation on binding of the TF to an inhibitor and to the gene's URS. Oscillations arise quite naturally for reasonable values of the kinetic parameters. (I replace the explicit time delay by an intermediate protein between the mRNA and the TF.) Other readers besides me might wonder why this standard approach is faulty, in the authors' opinion.

I would only insist on one clarifying correction. On lines 88-97, the authors acknowledge earlier demonstrations that competition-driven titration can create sufficient nonlinearity to generate self-sustained oscillations in negative-feedback circuits (referencing Kim & Forger #33 and three papers by Buchler #31, 32, 34). Then they say that "implementation is in all of these cases [35-40] based on a mechanistically incorrect mechanism: namely allosteric cooperative binding for which the Hill equation was derived." I have not checked publications [35-40], which may indeed rely on Hill function nonlinearities, but papers [31-34] are not 'mechanistically incorrect' in my opinion. (See also, the pdf I have attached to this review.) Maybe the authors mean to say something like this: "Previous authors have shown that competition-driven titration effects can lead to highly nonlinear (ultrasensitive) response curves [31, 32, 34] and to oscillations when embedded in a negative-feedback loop [33]. However, in many cases [35-40] such 'titration oscillators' are based on mechanistically incorrect assumptions, namely, allosteric cooperativity leading to Hill function nonlinearities. Kim and Tyson [41] level similar criticisms at the use of standard quasi-steady state approximations to derive Michaelis-Menten kinetics in complex networks of interacting proteins."

If what I say is correct, then the authors need to address how their approach improves on the analysis of competition-driven ultrasensitivity analyzed in [31-34].

NOTE: AN ATTACHMENT IS UPLOADED

Reviewer #2: In this paper Landman et al. use methods from statistical physics to develop a model of gene transcription. The major novelty in the approach lies in the fact that nonlinear functions that describe transcription rates are derived using statistical mechanics rather than by assuming functional forms a priori (e.g. Hill functions).

The manuscript is not written in an accessible enough manner. In my opinion, too much prior knowledge of statistical mechanics is assumed by the authors (given the nature of the PloS Comp Biological readership). The criticisms of Reviewer 1 in the previous submission have not been adequately addressed.

I found it difficult to formulate the proposed model in a manner that makes it comparable to the corresponding Hill function models. For example, in equation (18) Phi is a function of p whilst in equation 17 Phi is a function of many factors (e.g. free energies, fugacities). It has not been made clear how these are related. As many readers will not have a background in statistical mechanics, the interpretation and derivation of the fold change function (Phi) has not been made sufficiently clear.

There were numerous statements made that did not include appropriate citation

(e.g. transcription factors such as LacI tend to be poorly soluble in water, the binding free energy of RNA polymerase to its specific site is relatively weak …)

It is my view that this article, in its current formulation, does not fall within the remit of PloS Computational Biology. It is not clear what profound biological insights come from the work.

Reviewer #3: Oscillatory genetic networks are at the heart of circadian

rhythms. Understanding the conditions under which genetic networks can

oscillate is therefore an important biological question. It is well

known that cooperative binding of transcription factors to the DNA can

yield the required non-linearity for oscillations. However, titration

can also provide the required non-linearity. Yet, a thorough

theoretical description and analysis have been lacking. In their

manuscript, Landman and coworkers provide such a description and

analysis. They use their description to derive under which conditions

titration can induce oscillations. They show that the required

non-linearity can arise via the binding of transcription factors to

titration sites, by having multiple gene copies, and by having

auxiliary sites forming DNA loops. Moreover, the show what the minimum

number of binding sites and gene copies, and the maximum looping free

energy, are for obtaining oscillations. These are important results

that are of wide interest. The manuscript is also clearly written and

the figures are beautiful and clear. I therefore believe that

publication of this manuscript in PLCB is warranted.

While I like the efforts of the authors to apply their model to

existing experimental data (the Stricker data) , I also believe that

their work can be seen as predictions for new experiments. Figs. 5c-e

provide nice predictions that could be tested experimentally. In

particular, titration sites could be introduced into the genome.

My most important comment is:

- I am puzzled by the fluctuations in the delay. Are these dynamic

fluctuations of the delay in time? If so, on what timescale do the

delays fluctuate? Or are these fluctuations arising from

cell-to-cell variability, which are static on the timescale of the

oscillations? It is also not clear to me how these fluctuations are

accounted for in the stability analysis - via simulations (direct integration of DDE), or

via the theoretical stability analysis (but if so, how)?

Minor comments:

- P_i is not defined before or immediately below Eq. 4 (but only much later)

- Eq. 4 is presented as a self-consistency equation, but that it is

not directly from the equation itself (a self-consistency equation

has the variable typically on both sides of the equality, but this

is not immediately apparent here).

- Introduction, 2nd paragraph below Eq, 4, the interpretation of the

fraction θ(λ1,λ2,...) / θ(λ1,0,...) only becomes clear after the

reader is informed to which species 1, 2, etc. correspond to. It

would be good to iterate that RNAP corresponds to species 1, and

that the other indices refer to the transcription factors (TFs).

- I like the grand-canonical description, but in practice it is not

very different from the more conventional description developed by

Hwa and coworkers, where the partition function is written as a sum

of exponents of [x] / ΚD, where [x] is the concentration of the

transcription factor in the cytoplasm, which can be thought of as

the fugacity, and KD is given by e^{-βε). In both cases, one needs

to solve a constraint, which indeed arises from a mass

balance. Perhaps the authors could briefly comment on this

viewpoint.

- While I think for this work it is fine to assume that the mRNA and

protein degradation rates are equal, in many systems the mRNA

lifetime is shorter than the cell-division time, which means that

degradation is the dominant mechanism for decay, not dilution by

growth.

- I find it intuitive that titration sites can induce the required

non-linearity for generating oscillations. The result that just

having more than one gene copy is sufficient for inducing

oscillations is less intuitive to me (Fig. 2). What is the minimal

gene copy number at which oscillations arise? And is the behavior of

this system then highly similar if not identical to that of a single

gene but with an identical number of TF binding sites? Or does the

fact that each TF binding sites comes with a promoter and its

adjacent gene (which leads to the production of proteins) make a big

difference? I presume panels c and d of Fig. 5 give the answer - to

get the required sharpness of |dΦ/dp| ~ 4 οr higher, 20 competitor

sites would be required for a single gene, while a 100 copies of

identical genes would be required. So there is indeed a

difference. Is this interpretation correct?

- line 569: "such the real" -> "such that the real"

- Caption Fig. 5a: For ease of reading, please explain in the caption what is on the axes.

- line 611: "We show the positive branch of the real part of λ in

Fig. 5a". I think I see what the authors mean, but this sentence is

confusing, I feel. The contourplot is a contourplot of Im{λ}. The

white dashed line is the border between the oscillatory and the

non-oscillatory regime, which is the line where the real part of the

dominant eigenvalues λ equals 0 - this is clear. But what do the

authors mean by "positive branch of the real part" in this sentence (line 611)?

**Have the authors made all data and (if applicable) computational code underlying the findings in their manuscript fully available?**

Reviewer #1: Yes

Reviewer #2: Yes

Reviewer #3: Yes

PLOS authors have the option to publish the peer review history of their article (what does this mean?). If published, this will include your full peer review and any attached files.

Reviewer #1: **Yes: **John J. Tyson

Reviewer #2: No

Reviewer #3: **Yes: **Pieter Rein ten Wolde

Figure Files:

Data Requirements:

Reproducibility:

References:

---

## [Decision Letter · Decision Letter 1]

17 Aug 2023

Dear Mr. Landman,

Thank you very much for submitting your manuscript "Transcription factor competition facilitates self-sustained oscillations in single gene genetic circuits" for consideration at PLOS Computational Biology. As with all papers reviewed by the journal, your manuscript was reviewed by members of the editorial board and by several independent reviewers. The reviewers appreciated the attention to an important topic. Based on the reviews, we are likely to accept this manuscript for publication, providing that you modify the manuscript according to the review recommendations.

Sincerely,

Pedro Mendes, PhD

Section Editor

PLOS Computational Biology

Pedro Mendes

Section Editor

PLOS Computational Biology

Reviewer's Responses to Questions

**Comments to the Authors:**

Reviewer #1: See attachment

**Have the authors made all data and (if applicable) computational code underlying the findings in their manuscript fully available?**

Reviewer #1: Yes

PLOS authors have the option to publish the peer review history of their article (what does this mean?). If published, this will include your full peer review and any attached files.

Reviewer #1: **Yes: **John J. Tyson

Figure Files:

Data Requirements:

Reproducibility:

References:

---

## [Editor Report · Decision Letter 2]

18 Sep 2023

Dear Mr. Landman,

We are pleased to inform you that your manuscript 'Transcription factor competition facilitates self-sustained oscillations in single gene genetic circuits' has been provisionally accepted for publication in PLOS Computational Biology.

Best regards,

Pedro Mendes, PhD

Section Editor

PLOS Computational Biology

Pedro Mendes

Section Editor

PLOS Computational Biology

---

## [Editor Report · Acceptance letter]

22 Sep 2023

PCOMPBIOL-D-23-00261R2 

Transcription factor competition facilitates self-sustained oscillations in single gene genetic circuits

Dear Dr Landman,

I am pleased to inform you that your manuscript has been formally accepted for publication in PLOS Computational Biology. Your manuscript is now with our production department and you will be notified of the publication date in due course.

With kind regards,

Anita Estes
